# The First Case of Zika Virus Disease in Guinea: Description, Virus Isolation, Sequencing, and Seroprevalence in Local Population

**DOI:** 10.3390/v15081620

**Published:** 2023-07-25

**Authors:** Roman B. Bayandin, Marat T. Makenov, Sanaba Boumbaly, Olga A. Stukolova, Anastasia V. Gladysheva, Andrey V. Shipovalov, Maksim O. Skarnovich, Ousmane Camara, Aboubacar Hady Toure, Victor A. Svyatchenko, Alexander N. Shvalov, Vladimir A. Ternovoi, Mamadou Y. Boiro, Alexander P. Agafonov, Lyudmila S. Karan

**Affiliations:** 1State Research Center of Virology and Biotechnology «Vector», 630559 Kol’tsovo, Novosibirsk Oblast, Russiaskarnovich@vector.nsc.ru (M.O.S.);; 2Central Research Institute of Epidemiology, Novogireevskaya St. 3A, 111123 Moscow, Russia; 3Virology Research Center/Laboratory of Viral Hemorrhagic Fevers, Conakry, Guinea; 4Hôpital Régional de Faranah, Faranah, Guinea; 5Research Institute of Applied Biology of Guinea, Pastoria, CREMS, Kindia, Guinea

**Keywords:** Zika virus, seroprevalence, flavivirus, arbovirus, fever, mosquito-borne virus, vector-borne virus

## Abstract

The Zika virus (ZIKV) is a widespread mosquito-borne pathogen. Phylogenetically, two lineages of ZIKV are distinguished: African and Asian–American. The latter became the cause of the 2015–2016 pandemic, with severe consequences for newborns. In West African countries, the African lineage was found, but there is evidence of the emergence of the Asian–American lineage in Cape Verde and Angola. This highlights the need to not only monitor ZIKV but also sequence the isolates. In this article, we present a case report of Zika fever in a pregnant woman from Guinea identified in 2018. Viral RNA was detected through qRT-PCR in a serum sample. In addition, the seroconversion of anti-Zika IgM and IgG antibodies was detected in repeated blood samples. Subsequently, the virus was isolated from the C6/36 cell line. The detected ZIKV belonged to the African lineage, the Nigerian sublineage. The strains with the closest sequences were isolated from mosquitoes in Senegal in 2011 and 2015. In addition, we conducted the serological screening of 116 blood samples collected from patients presenting to the hospital of Faranah with fevers during the period 2018–2021. As a result, it was found that IgM-positive patients were identified each year and that the seroprevalence varied between 5.6% and 17.1%.

## 1. Introduction

Zika virus (ZIKV), a member of the family *Flaviviridae*, is a mosquito-borne virus that causes acute febrile illness in humans. Mosquitoes of the genus *Aedes* are the main vectors of ZIKV [1,2]. The virus was isolated from at least 16 different *Aedes* species [3]. Mosquitoes *Ae. aegypti* are the main vector in urban environments, while mosquitoes *Ae. albopictus* transmit the virus in both urban and rural areas [4,5,6].

There are three other routes of transmission for ZIKV: sexual transmission, vertical transmission (from mother to child), and blood transfusion [7,8,9]. ZIKV is the only flavivirus capable of crossing the placental barrier and infecting embryos or fetuses [10], which can lead to microcephaly and other fetal complications [11,12]. It was also shown that African and Asian strains differ in their ability to infect and lyse human placental cells [13]. Unlike Asian strains, African ZIKV strains cause severe lysis of placental cells [13], and, as the authors suggest, infection with African ZIKV strains in early pregnancy will lead to abortion but not to the development of fetal brain pathologies.

The virus was first isolated in 1947 from a rhesus macaque monkey (*Macaca mulatta*) in the Zika Forest of Uganda [14]. Thereafter, single cases/small outbreaks of human infection were reported in Uganda, Tanzania, Nigeria, Egypt, the Central African Republic, Côte d’Ivoire, Senegal, and Sierra Leone, as well as in India, Malaysia, the Philippines, Thailand, Vietnam, and Indonesia [15,16,17,18,19,20]. The first large outbreak of Zika virus disease was reported from the island of Yap in Micronesia in 2007 [21], followed by an even larger outbreak in French Polynesia in 2013 with more than 30,000 cases [22].

In 2015, ZIKV first appeared in South America (Brazil) [23,24], leading to an outbreak that spread across the countries of South and Central America and the Caribbean within a year, with a total of more than 4 million cases [25]. By 2017, cases of local Zika virus infection had been reported in 87 countries [26], while the main carriers of ZIKV, *Aedes* mosquitoes, are common in 64 more countries [27]. 

In West Africa, ZIKV was first isolated from *Ae. luteocephalus* collected in 1968 in the western part of Senegal [28]. Thereafter, ZIKV was detected in 1971 in Nigeria [29]. Serological and entomological data indicated ZIKV infections in Sierra Leone in 1972 [30], Gabon in 1975 [31], and Côte d’Ivoire in 1999 [32]. Close to Guinea, in Southeastern Senegal, more than 400 ZIKV strains have been isolated from mosquitoes, mainly from *Ae. africanus, Ae. luteocephalus, Ae. furcifer,* and *Ae. taylori* [28,33,34,35].

In Guinea, extensive virological studies of mosquitoes were carried out in 1978–1989 within the framework of a joint long-term project between Guinea and the USSR [36]. During this project, about 77,000 mosquitoes of 25 species were studied, including *Culex, Aedes, Eretmapodites, Uranotenia, Anopheles,* and *Ficalbia* [36]. In the collected mosquitoes, the authors found and isolated the following viruses: dengue, Wesselsbron virus, *Bunyamwera orthobunyavirus,* and *M’Poko orthobunyavirus* [36]. However, ZIKV has not been found in Guinea. 

Here, we report the first case of Zika virus infection in Guinea and provide virological and serological evidence of the presence of ZIKV in the local population.

## 2. Materials and Methods

### 2.1. Ethics

This study was conducted with the approval of the National Ethics Committee for Health Research of Guinea (document number 061/CNERS/15, 31.08.2015) and in adherence to the Declaration of Helsinki [37]. Enrolled patients were informed about the methods and objectives of the study and gave their consent to participate.

### 2.2. Sample Collection and Preparation

In total, 152 patients admitted to the hospital in Faranah with fevers of undefined origin were enrolled in this study. For 20 patients, blood samples were collected in May 2018; for 69 patients, blood was collected in August 2018; 30 blood samples were collected in February 2019; and 33 samples were collected in July 2021. Blood samples were collected in sterile tubes with 0.5 M EDTA and were then centrifuged at 160× *g* for 10 min to pellet erythrocytes, after which plasma samples containing leukocytes were centrifuged at 10,000× *g* for 10 min to pellet white blood cells. Then, we extracted the total RNA from 0.5 mL of obtained plasma by using an AmpliSens Magno-sorb Kit (Central Research Institute of Epidemiology, Moscow, Russia).

### 2.3. PCR and ELISA Diagnostics

We performed PCR screening for ZIKV by using a commercial qRT-PCR kit (AmpliSens Zika virus FL; Central Research Institute of Epidemiology, Moscow, Russia) according to the manufacturer’s instructions. Plasma samples were tested for ZIKV IgM and IgG antibodies using “AntiZika Virus ELISA (IgG/IgM)” reagent kits (Euroimmun AG, Lubeck, Germany).

### 2.4. Virus Isolation and Sequencing

C6/36 (ATC-15; CCL-126) cells (150–200 thousand cells in 1 mL) were cultured in a 25 cm^2^ cell culture flask (Biologix, Shandong, China) with DMEM/F12 (1:1) medium (Thermo Fisher Scientific, Waltham, MA, USA) supplemented with 5% fetal bovine serum (Thermo Fisher Scientific, Waltham, MA, USA) and 200 µg/mL of penicillin and streptomycin (1:1) (Thermo Fisher Scientific, Waltham, MA, USA). C6/36 cells were infected with 1 mL of a mixture of 100 µL of PCR-positive Zika virus plasma and 900 µL of DMEM F12 medium (1:1) (prefiltered through 0.45 µm) and were incubated for 1 h at 28 °C. They were then washed with phosphate-buffered saline and covered in DMEM/F12 with 2% fetal bovine serum. The increase in the concentration of the virus was controlled by RT-PCR. After reaching a plateau (10–12 days), an aliquot of the virus-containing liquid was taken for sequencing and subsequent passages. Further passages follow the same procedure. There were three passages in total. 

Complete genome Sanger sequencing of the ZIKV strain was performed with a panel of specific primers (Table A1). The amplified fragments were sequenced on an Abi 3130xl genetic analyzer according to the manufacturer’s instructions (Thermo Fisher Scientific, Waltham, MA, USA). The complete genome sequence was deposited in GenBank (MN025403).

### 2.5. Data Analysis

The nucleotide sequences were aligned using Mega X [38]. Phylogenetic analyses were performed based on nucleotide alignment using the maximum-likelihood method with the best-fit substitution model (GTR + G + I). Support for the tree was assessed with 1000 ultrafast bootstrap replicates [39]. The consensus trees were reconstructed with IQ-TREE v1.6.12 [40] and visualized in iTOL v6.6 (https://itol.embl.de, accessed date 27 December 2022).

## 3. Results

### 3.1. Case Description and Diagnostics

A total of 152 plasma samples were tested with respect to ZIKV through qRT-PCR. A plasma sample from a woman from Faranah yielded a positive signal with a Ct-value of 35.8. The sample was collected on 29 August 2018 from a 27-year-old pregnant woman (in the 16th week of gestation) who lived in the Dandaya District of Faranah. She claimed that the symptoms of the illness had begun 5–6 days before, including headache, weakness, and fever. She had been vaccinated against yellow fever. ELISA showed positive results with respect to IgM and negative results with respect to the IgG antibodies specified for the NS1 protein of ZIKV (Figure 1).

On 13 September (the 20th day after the onset of the disease), the patient was called to the hospital for re-examination. The medical examination showed no symptoms of infection. A repeat blood test was taken and showed PCR-positive results with respect to ZIKV, a negative serostatus with respect to the IgM antibodies specific to the NS1 protein, and seroconversion of the IgG antibodies specific to the NS1 protein of ZIKV. Unfortunately, after that, contact with the patient was lost, and we were unable to track the outcome of the pregnancy.

### 3.2. Serosurvey

Since ZIKV viremia mainly lasts a few days, we performed ELISA screening to detect IgM antibodies, which circulate in the blood for up to 12 weeks or longer after exposure. A total of 116 samples collected from patients presenting to the hospital with fevers were tested. We found evidence of the possible circulation of ZIKV in Faranah Prefecture, Guinea (Table 1). ZIKV IgM prevalence among febrile patients varied between 5.6% and 17.1% in different seasons, with an average prevalence of 14.7%. These data should be interpreted with caution, as they are based on samples collected only from febrile patients.

### 3.3. Phylogenetic Analysis

We infected the C6/36 cell culture with the blood plasma of the ZIKV-positive pregnant women from Faranah (sample dated 29 August 2018). Three passages were carried out, followed by qRT-PCR with a positive signal and a Ct-value of 14.3. The resulting Faranah/18 strain was sequenced. A phylogenetic analysis showed that the Faranah/18 strain belonged to the African lineage of ZIKV, and the closest sequences were the strains isolated from mosquito pools in 2011 and 2015 in Senegal (Figure 2). Within the African lineage, the Guinean isolate clustered together with the West African strains from Senegal and Nigeria, which form a separate subgroup and are distinct from the strains from Uganda and the Central African Republic.

### 3.4. Mutation Profile

The phylogenetic analysis showed that the widespread occurrence of ZIKV during the pandemic correlated with the occurrence of seven amino acid substitutions in the Asian strains [42] (Table 2):Capsid protein C coding region, position 106 (hereinafter, the position in the amino acid sequence of the ZIKV polyprotein relative to the beginning of the open reading frame is indicated, GenBank accession number of the reference: ASR91936), amino acid substitution: A or T;Membrane glycoprotein precursor prM coding region, position 123, amino acid substitution: V or A;Membrane glycoprotein precursor prM coding region, position 139, amino acid substitution: S or N;Membrane glycoprotein precursor prM coding region, position 143, amino acid substitution: E or K;Envelope protein E coding region, position 763, amino acid substitution: V or M;Nonstructural protein NS1 coding region, position 982, amino acid substitution: A or V;RNA-dependent RNA polymerase NS5 coding region, position 3392, amino acid substitution: M or V.

The ZIKV strain Faranah/18 (GenBank ID MN025403) contained in the listed positions the amino acids that are typical for the strains of the African lineage (Table 2). These isolates contained an alanine in position 106 (Capsid protein) and a valine in position 3392 (NS5 protein), which enhanced neurovirulence in mice [42]. In addition, African strains contained a valine in position 982 (NS1 protein), which enhanced viral replication in *Ae. aegypti* mosquitoes [43]. The alanine in position 123 and the lysine in position 143 increased infection and transmission efficiency by *Ae. aegypti* and/or enhanced replication in models for human infection [43,44].

A comparison of Guinean ZIKV with the postpandemic Asian and American strains showed that the main differences in the mutations prM-139, prM-143, and E-763 were as follows: In the American strains, in position 139 (prM coding region), serine (S) was replaced with asparagine (N), which enhanced neurovirulence in newborn mice, and valine (V) was replaced with methionine (M) in position 763 (E-gene), which enhanced fitness in nonhuman primates [42]. In position 143 (prM), the Guinean strain (like other African isolates) contained lysine, which made it more virulent than the American strains that had glutamic acid in this position [44]. The prepandemic Asian strains had the least dangerous mutation profile in this list and contained amino acids in positions 123 (prM), 143 (prM), 982 (NS1), and 3392 (NS5), which were inferior to all other strains in terms of neurovirulence in newborn mice and viral infection in *Ae. Aegypti* mosquitoes [42,43,45].

## 4. Discussion

In this paper, we describe the first confirmed case of Zika fever in Guinea: the virus was isolated from the blood sample of a pregnant woman who lived in Faranah. Unfortunately, we were not able to trace how the disease proceeded or what consequences it led to since communication with the patient was lost. Although this is the first documented case, it is likely that the Zika virus had previously circulated in Guinea and was not detected due to a lack of diagnostic tools, the predominantly mild course of the disease in humans, and the similarity of symptoms to malaria, which is endemic to this region. We suppose that the Zika virus has been circulating in Guinea for some time. There are several indirect pieces of evidence supporting this suggestion. For example, a number of West African countries are endemic for ZIKV: evidence of the virus’s presence or serological markers has been found in neighboring Senegal [35], Sierra Leone [30], Liberia [46], Côte d’Ivoire [32], Gabon [2], Mali [47], Burkina Faso [48], and Nigeria [29]. Furthermore, during the ELISA screening of patients with fevers for anti-Zika IgM in Faranah (Guinea), we found that, on average, 14.7% of patients were seropositive. IgM-positive patients were detected in Faranah from 2018 to 2021, indicating the continued circulation of ZIKV in this region for at least three years. This is supported by similar results from Senegal, where the long-term circulation of ZIKV has been proven through molecular and virological methods, and ZIKV IgM prevalence among febrile patients was between 5% and 7.5% [49]. It is important to note that, to some extent, our serological data should be interpreted with caution due to the possible nonspecific detection of antibodies specific to other flaviviruses circulating in Guinea. Although antibodies specific to the NS1 protein have been shown to be largely Zika-specific, antibodies specific to the E protein are more cross-reactive [50]. 

In favor of the assumption that the virus has probably been circulating in Guinea for quite a long time, it can be said that the Zika virus has been detected in neighboring Senegal for several decades [28]. The borders between countries for local residents are quite open. Infected people and mosquitoes can travel with vehicles throughout the countries. It has also been shown that some species of mosquitoes of the genus *Anopheles* move at altitudes of 40–290 m with a tailwind over considerable distances, up to 300 km per night [51]. At the same time, 80% of the captured mosquitoes were females, of which 90% fed before migration, which means that various pathogens also travel considerable distances. It is likely that mosquitoes of the genus *Aedes* can also travel considerable distances. The Faranah/18 strain was isolated in 2018 after the end of the pandemic, which was dominated by Asian/American strains of ZIKV. With ever-increasing passenger traffic, global trade relations have led to the migration of carriers of various infections. The outbreak of Zika fever occurred in the Cape Verde Islands in 2015–2016, with 7580 suspected cases. Despite the location of these islands being next to the West African coast, the sequencing showed that the outbreak was caused by the Asian strains of ZIKV [52]. The phylogenetic analysis indicated that the strains were probably imported from Brazil between June 2014 and August 2015. In Angola, investigations of cases of previously unseen microcephaly identified Asian ZIKV isolates that were probably imported from Brazil between July 2015 and June 2016 [53]. At the same time, in Brazil, African strains of ZIKV have been detected in nonhuman primates and mosquitoes in the southern and southeastern regions, which are more than 1500 km apart, indicating that African strains have spread quite widely in Brazil [54]. All this suggests an even more global nature to the spread of ZIKV than was thought during the pandemic. Nevertheless, in Guinea, we have identified a typical strain belonging to the African lineage of ZIKV. Perhaps this is due to the fact that Guinea and its neighboring countries are less involved in the processes of globalization, although other factors are not excluded. Phylogenetically, all known ZIKV sequences are divided into Asian and African lineages. The African line, in turn, is divided into two sublineages: Ugandan and Nigerian. The strain we identified belongs to the African lineage and the Nigerian sublineage. The Faranah/18 strain has the highest homology with Senegal 2011 and Senegal 2015. Homology in nucleotide and amino acid sequences with the Senegal 2011 strain was 98.60% and 99.92%, and that with the Senegal 2015 strain was 98.48% and 99.87%, respectively. In addition, our strain has high homology (98.17% and 99.56%) with the strain DakAr41667 (MF510857) isolated in Senegal in 1984. These data suggest that ZIKV has been circulating in West Africa, including Guinea, for quite some time. However, no outbreaks of ZIKV have been reported in this region.

Previous experimental studies have shown that African strains have higher transmissibility, viremia, and pathogenicity than Asian strains. African strains have significantly higher titers in cell cultures, a higher rate of infection and dissemination through *Ae. aegypti* mosquitoes [55], and higher mortality in outbred mice [56]. The infection of immunocompromised mice with African strains resulted in weight loss, higher titers, and lower survival compared with those infected with Asian strains. When mosquitoes are infected, the infectious viral particles of the African strains of ZIKV are detected in the saliva of mosquitoes much earlier than those of Asian strains, and at low doses of mosquito infection, the infectious viral particles of Asian strains are not detected at all [41]. Additionally, unlike Asian strains, African strains cause severe lysis of placental cells, which probably leads to the termination of early pregnancy rather than the development of fetal brain pathologies such as microcephaly [27]. This assumption is supported by fetal death in immunocompetent mice infected with African strains [41]. At the same time, modern Asian strains lead to a more severe course of the disease and its consequences than the Asian strains that circulated before the pandemic [43]. 

Currently, a number of mutations have been described that significantly increase the pathogenicity and contribute to the spread of ZIKV. The phylogenetic analysis has shown that the widespread occurrence of ZIKV during the pandemic correlates with the occurrence of seven amino acid substitutions in Asian strains: C-T106A, prM-V123A, prM-S139N, E-V763M, NS1-A982V, NS5-M2634V, and NS5-M3392V [42]. The C-T106A mutation significantly increases infectivity and the viral load in *Ae. aegypti* mosquitoes, in primary dendritic cells, monocytes, and human macrophages, and in ifnar1 −/− mice. The prM-V123A, NS1-A982V, and NS5-M3392V mutations also increase the infectivity of the virus in mice and mosquitoes [45], while the NS1-A982V mutation leads to the inhibition of interferon-beta production and, accordingly, to a more severe infection [57]. The prM-S139N mutation significantly increases infectivity in human and mouse neural progenitor cells and impairs the differentiation of these cells, leading to more severe microcephaly with a thinner cortex, greater apoptosis of the fetal brain cells of pregnant mice, and higher neonatal mouse mortality [58]. The E-V763M mutation significantly increases neurovirulence and the viral load in the brains of newborn CD-1 mice, increases transplacental transmission, and significantly increases their mortality [42]. The NS5-M2634V mutation does not have a significant effect on replication in various cell cultures as well as on pathogenesis and virulence in mice; nevertheless, this mutation is fixed in all modern Asian strains [59]. The prM-E143K mutation, characteristic of African strains, increases the cytopathic effect and the titers of intracellular and extracellular virions; allows much better attachment to the cell membrane and penetration into human cell lines TE617, SF268, and HMC3; and ensures the release of the virus [44].

The Faranah/18 strain, like most African strains, contains five of the seven mutations described above that increase their aggressiveness: C-T106A, prM-V123A, prM-E143K, NS1-A982V, and NS5-M3392V. The main difference among the already described mutations between African and modern Asian strains is the presence of the prM-E143K mutation and the absence of significant prM-S139N and E-M763V mutations. African strains cause a more severe course of the disease compared with modern Asian strains, and it can be assumed that the prM-E143K mutation is one of the main reasons for the properties of African strains. At the same time, it is likely that not all significant mutations have yet been identified and that the mutual influence of mutations has not yet been identified. C-T106A, prM-V123A, NS1-A982V, and NS5-M3392V mutations have been shown to increase ZIKV infectivity in mice and mosquitoes synergistically [45]. 

At the same time, the introduction of several significant mutations into the FSS13025 strain (C-T106A + E-V763M, C-T106A + NS5-M3392V, E-V763M + NS5-M3392V, and C-T106A + E-V763M + NS5-M3392V) did not significantly increase mortality rates and likely had an adverse effect on the E-V763M mutation. The FSS13025 E-V763M strain with a single substitution had the highest neurovirulence among all of the several mutation variants tested [42]. The introduction of a single prM-S139N mutation into the FSS13025 strain resulted in severe microcephaly in the fetus of pregnant mice [58], while the prM-N139S backmutation in the modern Asian strain PRVABC59 did not affect vertical transmission or microcephaly [60]. It should also be noted that a phylogenetic analysis that correlated seven mutations (C-T106A, prM-V123A, prM-S139N, E-V763M, NS1-A982V, NS5-M2634V, and NS5-M3392V) with the ZIKV pandemic was performed only for the coding parts of the genome of 73 strains and, accordingly, did not take into account the influence of the 5′ and 3′ ends on the level of replication, infectivity, virulence, and mortality.

Despite the higher aggressiveness of African strains, only isolated cases of infection are noted in African countries. One reason is the lack of diagnostic tools for Zika and other infections. This region is endemic for malaria and various febrile diseases, which, with their similar clinical symptoms, lead to incorrect diagnoses. Although African strains are more aggressive, severe cases are quite rare and probably only occur in large numbers in both African and Asian strains [61].

It is known that African strains lead to the severe lysis of placental cells [27] and fetal death in immunocompetent mice [41]. In a significant percentage of cases, high doses of African strains caused the death of the fetus of rhesus monkeys, but in cases of successful delivery, newborns did not differ significantly from newborns in the control group. At the same time, infected females did not have clinical symptoms in the form of a rash or fever [62]. It is possible that the lack of reported cases of microcephaly in Africa caused by African strains is associated with fetal death; meanwhile, in successful deliveries, no pathologies are recorded, and in the absence of fever and rash, an incorrect clinical diagnosis is made. Additionally, it cannot be ruled out that people simply do not seek medical help due to the mild course of the disease for economic or other reasons. It seems that for the emergence of an epidemic in Africa, the coincidence of many factors is necessary.

Given that the properties of African strains allow them to spread much faster and cause much more serious consequences for the health of people, especially pregnant women and the fetus, public health should be more serious about diagnosing the virus in patients with fevers of unknown etiology as well as controlling its spread.

## 5. Conclusions

It is likely that the Zika virus has been circulating in Guinea for at least 3 years. African strains are known to cause more harm to human health, especially to pregnant women and fetuses. Unfortunately, we were not able to trace the outcome of the disease in a pregnant patient since communication with her was lost.

The phylogenetic analysis showed that the resulting strain belonged to the African lineage and the Nigerian sublineage and that it had the highest homology with the Senegal 2011 and Senegal 2015 strains. The Faranah/18 strain had typical mutations that could lead to serious health consequences and could contribute to the wide spread of the virus. However, for reasons that are not fully understood, there have been no major outbreaks in Africa.

## Figures and Tables

**Figure 1 viruses-15-01620-f001:**
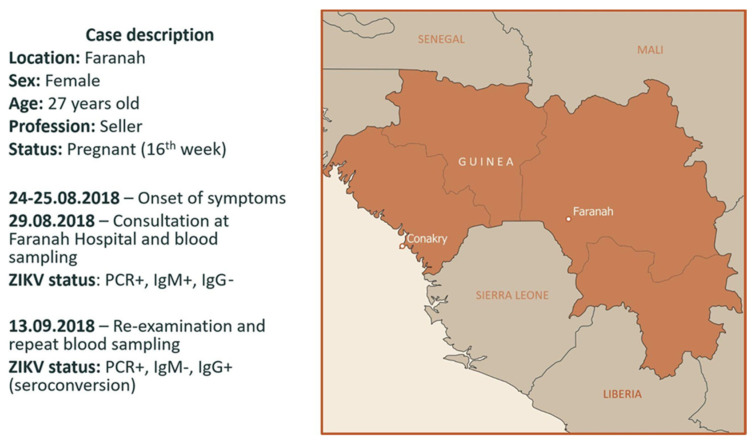
Timeline and case description of Zika virus disease in Faranah, Guinea.

**Figure 2 viruses-15-01620-f002:**
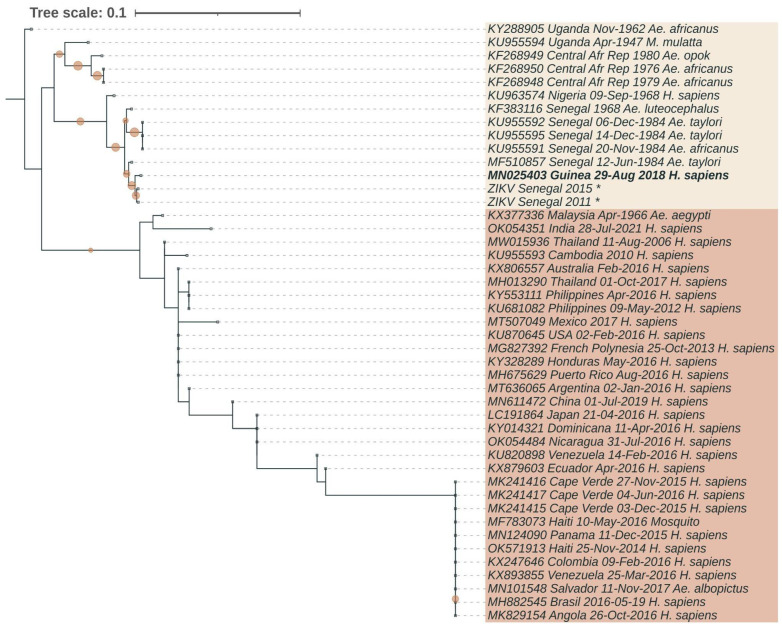
Phylogenetic position of ZIKV strain obtained in this study. The consensus tree was generated from 1000 ultrafast bootstrap replicate maximum-likelihood trees using a GTR + G + I nucleotide substitution model of the partial ZIKV polyprotein (10302 bp). The sequence generated during this study is indicated with bold text. The colored background represents the African lineage (sandy) and the Asian lineage (brown) of ZIKV. Two sequences from Senegal (marked with asterisks) are not present in GenBank and were kindly provided by Louis Lambrechts [41]. The tree was midpoint-rooted, and the root position was verified by the Spondweni virus outgroup. The scale bar represents the number of nucleotide substitutions/site. The filled circles on branches indicate ultrafast bootstrap values greater than 0.8.

**Table 1 viruses-15-01620-t001:** Results of RT-PCR screening for the presence of the ZIKV RNA in human blood and an ELISA assay for ZIKV IgM antibodies.

Collection Date		Season	Number of Samples				
			Studied through RT-PCR	RT-PCR Positive	Studied through ELISA	IgM Positive	IgM Positive, % (CI95%)
2018	May	Rainy season	20	0	18	1	5.6 (0.1–27.3)
2018	August	Rainy season	69	1	35	6	17.1 (6.6–33.6)
2019	February–March	Dry season	30	0	30	5	16.7 (5.6–34.7)
2021	July	Rainy season	33	0	33	5	15.2 (5.1–31.9)
Total			152	1	116	17	14.7 (8.8–22.4)

**Table 2 viruses-15-01620-t002:** Comparison of the most important mutations in the Guinean isolate of ZIKV with other African (yellow background), Asian (green background), and American strains (brown background).

GBank №/ Strain	Country	Date	Source	C-T106A	prM-V123A	prM-S139N	prM-E143K	E-V763M	NS1-A982V	NS5-M3392V
KU955594	Uganda	April 1947	*M. mulatta*	A	A	S	K	V	V	V
KU963574	Nigeria	9 September 1968	*H. sapiens*	A	A	S	K	V	V	V
KF268950	Central Afr. Republic	1976	*Ae. africanus*	A	A	S	K	V	V	V
MF510857	Senegal	12 June 1984	*Ae. taylori*	A	A	S	K	V	V	V
MK241415	Cape Verde	3 December 2015	*H. sapiens*	A	A	N	E	M	V	V
Senegal 2011	Senegal	2011	Mosquito pools	A	A	S	K	V	V	V
Senegal 2015	Senegal	2015	Mosquito pools	A	A	S	K	V	V	V
MN025403	Guinea	29 August 2018	*H. sapiens*	A	A	S	K	V	V	V
KX377336	Malaysia	April 1966	*Ae. aegypti*	A	V	S	K	V	A	V
MW015936	Thailand	11 August 2006	*H. sapiens*	T	V	S	E	V	A	M
KU955593	Cambodia	2010	*H. sapiens*	T	V	S	E	V	A	M
MG827392	Fr. Polynesia	25 October 2013	*H. sapiens*	A	A	N	E	M	V	V
MH013290	Thailand	1 October 2017	*H. sapiens*	A	A	S	E	M	V	V
OK054351	India	28 July 2021	*H. sapiens*	A	V	S	E	V	I	V
AMK49492	Indonesia	30 December 2014	*H. sapiens*	T	A	S	E	M	V	V
OK571913	Haiti	25 November 2014	*H. sapiens*	A	A	N	E	M	V	V
MN124090	Panama	11 December 2015	*H. sapiens*	A	A	N	E	M	V	V
MH882545	Brasil	19 May 2016	*H. sapiens*	A	A	N	E	M	V	V

## Data Availability

Data sharing is not applicable for this article.

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
