# Peer review of "The First Case of Zika Virus Disease in Guinea: Description, Virus Isolation, Sequencing, and Seroprevalence in Local Population"

_viruses, 2023, doi:10.3390/v15081620_

Round 1

Reviewer 1 Report

Introduction of the paper is not in flow. Paras 1 & 2 cover the epidemiology, para 3 is vector, para 4 is transmission routes and para 5,6 is again epidemiology of zika.

Unite epidemiology paragraphs, following by vector and transmission routes.

Sequencing was done by Sanger methods using ABI 3130 xl genetic analyzer, Which gives a good read of 800-900 base pairs long sequence. Table A1 shows the primers and length of the sequence amplified by these primers. How these primers were designed?

One of the sequences amplified is of 1051 bp length, which is a longer fragment than the efficiency of sequence read by the sequencer, why? It was better to split it into smaller fragments.

After sequencing, authors compared the sequence of current zika with reported sequences and especially studies the key mutations, which are already reported in the previous studies.

If there will be inclusion of an experience bioinformatician in the team, author can analyze the new mutations in the zika genome, which may affect the virus transmission or replication rates, as several new variants of covid-19 are reported, with different transmission and replication efficiencies. You may include this in your future studies. In this study, sequence comparison in ok.

Discussion part is well written.

Conclusion

First 6.5 lines of conclusion is comprising of methodology. It is not conclusion. Remove them.

References: some of the references are more than 50 years old. Try to include maximum new references. These information are available in new papers.

Minor English editing requried

Author Response

Dear Reviewer!

We are pleased to have been given the opportunity to revise our manuscript “The first case of Zika virus disease in Guinea: description, virus isolation, sequencing and seroprevalence in local population” which we are submitting as a Research Article in Viruses Journal. We have carefully considered the comments offered by the reviewers. Here, we explain how we revised the paper based on their comments and recommendations.

We hope that these revisions improve the paper. Next, we offer detailed responses to the reviewers’ comments.

Responses to Reviewer #1’s comments:

  1. Comment:

Introduction of the paper is not in flow. Paras 1 & 2 cover the epidemiology, para 3 is vector, para 4 is transmission routes and para 5,6 is again epidemiology of Zika. Unite epidemiology paragraphs, following by vector and transmission routes.

RESPONSE: Thank you for this comment. We have combined the paragraphs on epidemiology into one paragraph. Lines 40-50.

  1. Comment:

Sequencing was done by Sanger methods using ABI 3130 xl genetic analyzer, Which gives a good read of 800-900 base pairs long sequence. Table A1 shows the primers and length of the sequence amplified by these primers. How these primers were designed? One of the sequences amplified is of 1051 bp length, which is a longer fragment than the efficiency of sequence read by the sequencer, why? It was better to split it into smaller fragments.

RESPONSE: The primers were designed for versatility so that different Zika virus genetic lineages could be sequenced with these primers. In this regard, when constructing the alignment, the most conservative areas were selected, which were sometimes located at a rather large distance from each other. At the same time, we sequenced in the forward and reverse direction, which confirmed the correctness of sequencing. In those places where there was only a direct or only reverse sequence, we sequenced three times and the results coincided.

  1. Comment:

After sequencing, authors compared the sequence of current Zika with reported sequences and especially studies the key mutations, which are already reported in the previous studies. If there will be inclusion of an experience bioinformatician in the team, author can analyze the new mutations in the Zika genome, which may affect the virus transmission or replication rates, as several new variants of covid-19 are reported, with different transmission and replication efficiencies. You may include this in your future studies. In this study, sequence comparison in ok.

RESPONSE: Thank you for your comment. In the future, we plan to study the effect of mutations on the rate of replication and the severity of the course of diseases using virological and bioinformatic methods.

  1. Comment:

Conclusion. First 6.5 lines of conclusion is comprising of methodology. It is not conclusion. Remove them.

RESPONSE: Thanks for your comment, we have changed the Conclusion section.

Lines 367-377.

  1. Comment:

References: some of the references are more than 50 years old. Try to include maximum new references. These information are available in new papers.

RESPONSE: Thank you for your valuable comment. Dr. Dick has two articles in the same journal in 1952. One article about the discovery of the Zika virus, the second about the pathogenicity of the Zika virus for humans, unfortunately, we mixed up these articles. We removed the article on human pathogenicity, but inserted an article on the discovery of the virus. We provide data for this and other articles because they contain data on the very first cases of detection of the Zika virus (or virus markers) in different regions of Africa and Southeast Asia. We believe that these data are important from the point of view of temporal and geographical distribution and that it is necessary to cite the primary sources and the works of those authors who discovered them. For example, the Zika virus was discovered in 1947 in Uganda, and in the early 1960s, virus markers were already detected in Southeast Asia (mutations occurred in the genome that reduced the spread and severity of the disease), which is confirmed by phylogenetic analysis. For this reason, we believe that it is desirable to leave these links in the article.       

Lines 428-429

  1. Comment:

Minor English editing requried

RESPONSE: Thank you for this suggestion. We had sent the manuscript for proofreading to the MDPI English Editing Service. We hope that the proofreading of the revised manuscript has improved the English of our article.

Translator        

Reviewer 2 Report

In this manuscript, Dr. Roman B. Bayandin and other authors reported a case report of Zika fever in a pregnant woman from Guinea, identified in 2018. They confirmed ZIKV viral RNA in this case by ZIKV-specific qRT-PCR and anti-Zika IgM and IgG antibodies via ZIKV-specific ELISA. They further isolated the virus on C6/36 cells and determined the consensus sequences of the isolated virus by Sanger sequencing. They performed a phylogenetic analysis and determined this isolated virus belong to the Nigerian sublineage of African lineage, close to ZIKV isolated from mosquitoes in Senegal in 2011 and 2015. In addition, they conducted serological screening of blood samples collected from patients with fever presenting to the hospital of Faranah during 2018-2021. They found that ZIKV IgM-positive patients occurred annually.  The study provides timely and essential information on the ZIKV African lineage-associated human case in the ZIKV field. The manuscript was organized well. I would suggest the following points that may further strengthen the manuscript.

1.       Line 145-147: “ZIKV IgM prevalence among febrile patients varied between 5.6% and 17.1% in different seasons with an average prevalence of 14.7%. These data should be interpreted with caution, as they are based on the samples collected only from febrile patients” demonstrated that ZIKV infection occurred annually in Guinea. It may be important to use a more strain-specific approach (RT-PCR or next-generation sequencing) to determine the circulated ZIKV strain in local mosquitoes and communities. The authors may add it as additional discussion or conclusion points to call for more public attention on 1) the potential circulation and adaptation of ZIKV African lineage in Africa; 2) closer monitoring of the outcome of the disease caused by ZIKV African strains in pregnant women.

2.       Line 104: C6/36 (ATCC; CCL-81) cells: please correct the catalog number of C6/36 cells. CCL-81 refers to Vero cells.

3.       Lines 107-111: The Ct-value of the original plasma samples was 35.8, suggesting a low abundance of the virus. Please provide more details regarding the isolation procedure that could help the audience understand how the virus was isolated, including how many cells were used, what type of plastic labware, and the volume of the initial input virus and inoculum during passages. 

4. Line 151: please clarify whether this is the same patient as described in lines 125-131. 

Author Response

We are pleased to have been given the opportunity to revise our manuscript “The first case of Zika virus disease in Guinea: description, virus isolation, sequencing and seroprevalence in local population” which we are submitting as a Research Article in Viruses Journal. We have carefully considered the comments offered by the reviewers. Here, we explain how we revised the paper based on their comments and recommendations.

We hope that these revisions improve the paper. Next, we offer detailed responses to the reviewers’ comments.

Responses to Reviewer #2’s comments:

  1. Comment:

Line 145-147: “ZIKV IgM prevalence among febrile patients varied between 5.6% and 17.1% in different seasons with an average prevalence of 14.7%. These data should be interpreted with caution, as they are based on the samples collected only from febrile patients” demonstrated that ZIKV infection occurred annually in Guinea. It may be important to use a more strain-specific approach (RT-PCR or next-generation sequencing) to determine the circulated ZIKV strain in local mosquitoes and communities. The authors may add it as additional discussion or conclusion points to call for more public attention on 1) the potential circulation and adaptation of ZIKV African lineage in Africa; 2) closer monitoring of the outcome of the disease caused by ZIKV African strains in pregnant women.

RESPONSE: Thank you for your comment. We have added to the discussion points on circulation and drawing public health attention to the diagnosis, control and health consequences of patients with fevers of unclear etiology, especially pregnant women and their future children.

Lines 356-359

  1. Comment:

Line 104: C6/36 (ATCC; CCL-81) cells: please correct the catalog number of C6/36 cells. CCL-81 refers to Vero cells.

RESPONSE: Thanks for the comment, we have changed the catalog number of the cell line to ATC-15; CL-126. Line 101

  1. Comment:

Lines 107-111: The Ct-value of the original plasma samples was 35.8, suggesting a low abundance of the virus. Please provide more details regarding the isolation procedure that could help the audience understand how the virus was isolated, including how many cells were used, what type of plastic labware, and the volume of the initial input virus and inoculum during passages.

RESPONSE: Thank you for your comment. We have described in more detail the method by which the Zika virus was isolated. Lines 101-114

  1. Comment:

Line 151: please clarify whether this is the same patient as described in lines 125-131.

RESPONSE: Thank you for this suggestion. We have added the folowing clarification to the text (Line 151):

We infected the C6/36 cell culture with the blood plasma of the ZIKV-positive pregnant women from Faranah (sample dated 08/29/2018). Lines 157-158.

We appreciate all your insightful comments. Thank you for taking the time and effort to help us improve our paper.

Round 2

Reviewer 1 Report

Reviewer's comments are addressed. Paper is acceptable in current format.